# Morphology, DNA Phylogeny, and Pathogenicity of *Wilsonomyces carpophilus* Isolate Causing Shot-Hole Disease of *Prunus divaricata* and *Prunus armeniaca* in Wild-Fruit Forest of Western Tianshan Mountains, China

**Shuanghua Ye [1], Haiying Jia [1], Guifang Cai [2], Chengming Tian [3] and Rong Ma [1,4,*]**

[1] College of Forestry and Horticulture, Xinjiang Agricultural University, 830052 Urumqi, China; 15894618315@sina.cn (S.Y.); xjaumr@sina.com (R.M.); jiahaiying0101@sina.com (H.J.)

[2] Forestry Technology Extension Center of Changji Prefecture, 831100 Changji, China; 13669949008m@sina.cn

[3] The Key Laboratory for Silviculture and Conservation of Ministry of Education, Beijing Forestry University, 100083 Beijing, China; chengmt@bjfu.edu.cn

[4] CAS Key Laboratory of Biogeography and Bioresource in Arid Land, Xinjiang Institute of Ecology and Geography, 830052 Urumqi, China

[*] Correspondence: xjaumr@sina.com; Tel.: +86-136-9935-5593

**Abstract:** *Prunus divaricata* and *Prunus armeniaca* are important wild fruit trees that grow in part of the Western Tianshan Mountains in Central Asia, and they have been listed as endangered species in China. Shot-hole disease of stone fruits has become a major threat in the wild-fruit forest of the Western Tianshan Mountains. Twenty-five isolates were selected from diseased *P. divaricata* and *P. armeniaca*. According to the morphological characteristics of the culture, the 25 isolates were divided into eight morphological groups. Conidia were spindle-shaped, with ovate apical cells and truncated basal cells, with the majority of conidia comprising 3–4 septa, and the conidia had the same shape and color in morphological groups. Based on morphological and cultural characteristics and multilocus analysis using the internal transcribed spacer (ITS) region, partial large subunit (LSU) nuclear ribosomal RNA (nrRNA) gene, and the translation elongation factor 1-alpha (tef1) gene, the fungus was identified as *Wilsonomyces carpophilus*. The 25 *W. carpophilus* isolates had high genetic diversity in phylogenetic analysis, and the morphological groups did not correspond to phylogenetic groups. The pathogenicity of all *W. carpophilus* isolates was confirmed by inoculating healthy *P. divaricata* and *P. armeniaca* leaves and fruits. The pathogen was re-isolated from all inoculated tissues, thereby fulfilling Koch's postulates. There were no significant differences in the pathogenicity of different isolates inoculated on *P. armeniaca* and *P. divaricata* leaves ($p > 0.05$). On fruit, G053 7m3 and G052 5m2 showed significant differences in inoculation on *P. armeniaca*, and G010 5m2 showed extremely significant differences with G004 7m2 and G004 5m2 on *P. divaricata* ($p < 0.05$). This is the first report on shot-hole disease of *P. armeniaca* (wild apricot) leaves and *P. divaricata* induced by *W. carpophilus* in China.

**Keywords:** *Wilsonomyces carpophilus*; *Prunus divaricata*; *Prunus armeniaca* (wild apricot); shot-hole disease; wild-fruit forest; China

## 1. Introduction

The Ili River Valley of the Western Tianshan Mountains is host to an important wild-fruit forest in Central Asia. The region has a great wealth of native plant species and has been considered an evolutionary center for fruit trees. Among them, *Prunus divaricata* and *Prunus armeniaca* are listed as endangered and protected plant species according to the regulations of the People's Republic of China [1,2].

*Prunus divaricata* belongs to the section *Prunus*, subgenus *Prunus*, within the subfamily Prunoideae of the family Rosaceae; it is often called wild cherry plum [3,4]. There are many types of cherry plums, collectively known to botanists as *P. cerasifera* Ehrh, which include horticultural and wild varieties. *P. cerasifera* Ehrh is usually used to refer to as the cultivated form of cherry plum, also known as myrobalan plum [5]. *P. divaricata* is used to characterize the wild population of cherry plum, which is distributed mainly in the Tianshan Mountains of Central Asia, the Caucasus, the Turkmen Mountains, Pamir Alai, Asia Minor, and the Balkans [6]. In China, wild cherry plums are only distributed in Huocheng County, Ili Kazak Autonomous Prefecture of Xinjiang Uygur Autonomous Region [5].

*Prunus armeniaca* Lam. (wild apricot) belongs to the genus *Prunus*, in the subfamily Prunoideae of the family Rosaceae [7]. The wild apricot of Central Asia is the oldest of six ecological geographical groups in the world and has played an important role in the origin and evolution of cultivated apricots [1]. The wild apricot is mainly distributed from the border between China and Kazakhstan, south to Kashmir, and west to Afghanistan in the Tianshan area of Central Asia [1]. In China, wild apricot is mainly concentrated in the valleys of Ili Xinyuan, Yining, Huocheng, and Gongliu Counties of Xinjiang [8,9].

However, due to increased human activity in these areas, the stability of the natural ecosystem has recently been jeopardized. In particular, the emergence of pests and diseases has limited the natural aging of trees and accelerated the decline of tree communities. The main disease in the wild-fruit forest of the Western Tianshan Mountains is canker disease caused by *Cytospora* and bacterial shot-hole disease caused by *Xanthomonas arboricola* pv. *pruni*, according to previous reports [1]. During August 2017 and 2018, preliminary observations found serious fungal shot-hole disease occurring on *P. divaricata* and *P. armeniaca*. The symptoms on the leaves include small circular purple lesions with pale centers, which gradually enlarged and became necrotic in the center until the center fell out, giving the appearance of shot-hole. Leaf photosynthesis will be seriously affected. On the fruits, the pathogen caused sunken necrotic lesions with purplish halos [10]. The growth of seeds and the regeneration of *Prunus divaricata* population also can be affected. These symptoms are similar to those of bacterial shot-hole disease. Of serious concern is the decline of the ancient *P. divaricata* forests and loss of natural vegetation renewal processes, which have threatened the forest's long-term health [11,12]. Defining the main plant pathogens associated with these native plant communities has become crucial to develop strategies for conserving and protecting such valuable ecosystems.

Fungal shot-hole disease, mainly caused by *Wilsonomyces carpophilus*, is another potentially important disease of stone fruits in the world. The leaves, fruits, twigs, dormant buds, and flower calyxes can be affected [13,14]. The disease was first reported in France and subsequently was found in South and North America, Africa, Australia, New Zealand, and Iran [15,16]. It affects a wide range of *Prunus* species, such as apricot, peach, nectarine, plum, cherry, and almond [13,17]. In China, the main hosts of fungal shot-hole disease include *P. armeniaca*, *P. campanulata*, *P. cerasus*, *P. domestica*, *P. mume*, *P. mandshurica*, *P. persica*, and *P. salicina* [10,16,18–20]. Cases have been recorded on *P. divaricata* only in California and Poland (https://nt.ars-grin.gov/ fungaldatabas-es/); it has not been recorded in China.

Shot-hole disease of stone fruits is caused by *Wilsonomyces carpophilus* (Lev.) Adask. (Bas. *Helminthosporium carpophilum*. syn. *Stigmina carpophila*, *Coryneum beijerinkii*, *Clasterosporium carpophilum*, *Thyrostroma carpophilum*, *Sciniatosporium carpophilum*, and *Sporocadus carpophilus*), an anamorph of the genus *Wilsonomyces* [21–23]. *Wilsonomyces* belongs to the Dothideomycetes, Pleosporomycetidae, Pleosporales, Dothidotthiaceae classification [24]. The taxonomy of the genus has been controversial, and Sutton regarded *Wilsonomyces* as a synonym of *Thyrostroma* [25].

However, recent phylogenetic analyses based on large subunit (LSU) nuclear ribosomal RNA, internal transcribed spacer (ITS), and translation elongation factor 1-alpha (tef1) sequences supported *Wilsonomyces* as representing a distinct genus and its location in the Dothidotthiaceae [24]. The present study followed the classification of *W. carpophilus* as proposed by Marin-Felix et al. [24]. Conidia of *W. carpophilus* are initially fusiform, aseptate, and hyaline [26]. As conidia mature, they are delimited from conidiogenous cells by a single transverse septum, and multiple (3–7) transverse septa subsequently form as the conidia separate from the conidiogenous cells [26]. Shot-hole disease is most harmful under cool and wet spring conditions, although it can occur and cause damage at any time during the growing season following prolonged wet weather [27,28].

In China, *W. carpophilus* has been recorded in 11 provinces and autonomous regions: Jilin, Hebei, Sichuan, Gansu, Henan, Jiangsu, Anhui, Guangdong, and Hubei Provinces, and Ningxia Hui and Xinjiang Uygur Autonomous Regions [10,16,18–20]. However, there has been very little research on the morphology, DNA phylogeny, and pathogenicity of *W. carpophilus* associated diseases in China.

The aim of the present study was to investigate the cause of shot-hole disease associated with *P. divaricata* and *P. armeniaca* in different regions of the Ili wild-fruit forest in Xinjiang. Following surveys, morphological studies and DNA phylogenies were used to identify the disease causal agent. Moreover, pathogenicity studies were performed to determine the virulence of diverse fungal isolates in *P. divaricata* and *P. armeniaca* and confirm Koch's postulates.

## 2. Materials and Methods

### 2.1. Shot-Hole Disease Investigation

In August 2017 and 2018, a survey for *P. armeniaca* shot-hole disease was conducted in the wild-fruit forest in four counties of Ili Kazak Autonomous Prefecture, Xinjiang Uygur Autonomous Region. A sample plot was randomly chosen in each county: Xinyuan County (43°22′03″–43°23′05″ N; 83°35′40″–83°37′00″ E; altitude: 1273–1446 m), Yining County (44°06′05″–44°11′24″ N; 81°36′27′- 81°61′28″ E; altitude: 1030–1163 m), Huocheng County (44°16′01″–44°26′21″ N; 80°41′20″–80°48′06″ E; altitude: 813–1326 m), and Gongliu County (43°11′08″–43°20′38″ N; 82°17′52″–82°45′14″ E; altitude: 1192–1411 m). Twenty trees were selected from each sample plot according to the five-point sampling method and four trees were randomly selected for investigation within a circular plot (r = 10 m), and then each tree was divided into east, south, west, and north directions, and 30 leaves and 10 fruits in each direction were taken [29,30].

*P. divaricata* fruits were investigated in Huocheng County (44°24′10.37″–44°26′19.04″ N; 80°43′09.09″–80°48′28.87″ E; altitude: 1164–1586 m). The survey method was the same as that used for *P. armeniaca*.

### 2.2. Sampling and Isolation

Eight fruit samples from *P. divaricata*, and 14 fruit and 25 leaf samples from *P. armeniaca* were collected and brought to the laboratory. Dry specimens of infected leaves were deposited in the herbarium of the Forest Pathology Laboratory of Xinjiang Agricultural University. In order to isolate the disease causal agent, margins of infected fruit and leaf lesions were cut into small pieces (5 × 5 mm), the surface was disinfected by immersion in 75% ethanol for 30 s, followed by 3% NaClO for 5 min, then the samples were rinsed twice in sterile distilled water and dried in sterilized Petri dishes. Five pieces of diseased tissue from each symptomatic fruit and leaf sample were placed in a 90 mm Petri dish filled with full-strength potato dextrose agar (PDA) medium (39 g/L; Sangon Biotech). Plates were incubated at 25 °C under a 12 h photoperiod in a light incubator (GX-260A; Ningbo Southeast Instrument Co. Ltd.) for 6 days. Mycelial fragments taken from the growing colony margin were transferred to fresh PDA [31–33].

For single-spore isolation, conidia were scraped off with a sterile needle and suspended in 1 mL sterile distilled water with 0.1% Tween 20. An aliquot of 50 mL conidial suspension was spread on water agar in a Petri dish. After incubation at 25 °C for 24 h, single germinated conidia were transferred under a stereomicroscope to PDA plates and incubated at 25 °C for another 36 h for

mycelium development [34]. Single-spore colonies were cultured on fresh PDA and pure cultures were stored in 15% glycerol at –80 °C [35,36]. Isolation plates were also checked regularly for the presence of the bacterial pathogen *Xanthomonas arboricola* pv. *pruni*.

## 2.3. Morphological Identification

Twenty-five isolates representing various morphological groups (three from *P. divaricata* fruits, 10 from *P. armeniaca* fruits, and 12 from *P. armeniaca* leaves) were used for morphological identification. Isolate morphology was assessed after 6 and 15 days of growth on PDA medium at 25 °C under the 12 h photoperiod. Anamorph characteristics, including the size and shape of conidiophores and conidia, were observed in water on a glass slide using an Olympus compound microscope (BX 53; Dongguan Yijiang Instrument Co. Ltd., Guangdong, China) [37]. Measurements of 50 conidia, including length and width, and 95% confidence intervals together with extreme values were determined [24].

The morphology of the 25 isolates growing on PDA medium was determined (Table 1). Mycelial plugs (5 mm in diameter) of the fungus growing on PDA were placed in the center of fresh PDA plates and incubated at 25 °C. Colony characteristics (color, concentric ring pattern, margin, presence of reproductive structures) were recorded. Mycelial growth rate was determined using three isolate replicates [38,39].

**Table 1.** *Wilsonomyces carpophilus* isolates collected and used in this study.

| Isolate no. | Host | Substrate | Location | GenBank Accession Numbers[2] | | |
| --- | --- | --- | --- | --- | --- | --- |
| | | | | ITS | LSU | tef1 |
| XJAU Y035 5m1 | *Prunus armeniaca* | Leaf | Huocheng | MN817623 | MN817648 | MN817598 |
| XJAU Y037 7m2 | *P. armeniaca* | Leaf | Xinyuan | MN817624 | MN817649 | MN817599 |
| XJAU Y038 7m2 | *P. armeniaca* | Leaf | Xinyuan | MN817625 | MN944917 | MN817600 |
| XJAU Y039 3m3 | *P. armeniaca* | Leaf | Xinyuan | MN817626 | MN817650 | MN817601 |
| XJAU Y040 7m2 | *P. armeniaca* | Leaf | Xinyuan | MN817627 | MN817651 | MN817602 |
| XJAU Y043 7m1 | *P. armeniaca* | Leaf | Yining | MN817628 | MN817652 | MN817603 |
| XJAU Y045 5m2-2 | *P. armeniaca* | Leaf | Yining | MN817629 | MN817653 | MN817604 |
| XJAU Y046 7m2 | *P. armeniaca* | Leaf | Yining | MN817630 | MN944918 | MN817605 |
| XJAU Y048 5m2 | *P. armeniaca* | Leaf | Huocheng | MN817631 | MN817654 | MN817606 |
| XJAU Y049 7m1 | *P. armeniaca* | Leaf | Huocheng | MN817632 | MN817655 | MN817607 |
| XJAU Y052 7m1 | *P. armeniaca* | Leaf | Huocheng | MN817633 | MN817656 | MN817608 |
| XJAU Y057 7m3 | *P. armeniaca* | Leaf | Gongliu | MN817634 | MN817657 | MN817609 |
| XJAU G048 3m3 | *P. armeniaca* | Fruit | Huocheng | MN817613 | MN817638 | MN817588 |
| XJAU G048 5m2 | *P. armeniaca* | Fruit | Huocheng | MN817614 | MN817639 | MN817589 |
| XJAU G048 5m3 | *P. armeniaca* | Fruit | Huocheng | MN817615 | MN817640 | MN817590 |
| XJAU G048 7m1 | *P. armeniaca* | Fruit | Huocheng | MN817616 | MN817641 | MN817591 |
| XJAU G049 7m1 | *P. armeniaca* | Fruit | Huocheng | MN817617 | MN817642 | MN817592 |
| XJAU G052 5m2 | *P. armeniaca* | Fruit | Huocheng | MN817618 | MN817643 | MN817593 |
| XJAU G052 5m3 | *P. armeniaca* | Fruit | Huocheng | MN817619 | MN817644 | MN817594 |
| XJAU G053 5m1 | *P. armeniaca* | Fruit | Huocheng | MN817620 | MN817645 | MN817595 |
| XJAU G053 7m3 | *P. armeniaca* | Fruit | Huocheng | MN817621 | MN817646 | MN817596 |
| XJAU G059 5m2 | *P. armeniaca* | Fruit | Xinyuan | MN817622 | MN817647 | MN817597 |
| XJAU G004 5m2 | *Prunus divaricata* | Fruit | Huocheng | MN817610 | MN817635 | MN817585 |
| XJAU G004 7m2 | *P. divaricata* | Fruit | Huocheng | MN817611 | MN817636 | MN817586 |
| XJAU G010 5m2 | *P. divaricata* | Fruit | Huocheng | MN817612 | MN817637 | MN817587 |

Location: Ili Kazak Autonomous Prefecture, Xinjiang Uygur Autonomous Region, China. ITS, internal transcribed spacers and intervening 5.8S nrDNA; LSU, 28S large subunit RNA gene; tef1, partial translation elongation factor 1-alpha gene.

## 2.4. DNA Extraction, PCR Amplification, and Sequencing

Mycelial discs of the 25 isolates were transferred to PDA plates covered with sterile cellophane and incubated at 25 °C in a 12 h photoperiod for 6 d [40]. Total DNA was extracted using the modified CTAB (cetyl trimethyl ammonium bromide; known as cetrimonium bromide) method [41]. Extracted DNA was checked for quality using 1% agarose gels [42]. Three pairs of primers were used for polymerase chain reaction (PCR) amplification. Partial gene sequences were determined for the internal transcribed spacer (ITS) and intervening 5.8S gene region, in order to identify the species involved. Additionally, partial large subunit (LSU) nuclear ribosomal RNA (nrRNA) gene and translation elongation factor 1-$\alpha$ (tef1) gene region were sequenced for some isolates to better resolve their identification [38]. Primers used were ITS1 (5'- TCC GTA GGT GAA CCT GCG G -3') and ITS4 (5'- TCC TCC GCT TAT TGA TAT GC -3') for ITS [43], NL1 (5'- GCA TAT CAA TAA GCG GAG GAA AAG -3') and NL4 (5'- GGT CCG TGT TTC AAG ACG G -3') for LSU [44], and EF1-688F (5'- CGG TCA CTT GAT CTA CAA GTG C -3') and EF1-1251R (5'- CCT CGA ACT CAC CAG TAC CG -3') for tef1 [45].

Conditions for PCR amplification were as follows: for ITS region: 94 °C for 2 min, followed by 35 cycles, where each cycle included 30 s at 94 °C, 30 s at 51 °C, and 40 s at 72 °C, then a final elongation step for 10 min at 72 °C. For LSU region: 2 min initial denaturation at 95 °C, followed by 35 cycles of 45 s denaturation at 95 °C, 45 s primer annealing at 55 °C, 1 min extension at 72 °C, and a final 10 min extension at 72°C. For tef1 region: 8 min initial denaturation at 95 °C, followed by 35 cycles of 15 s denaturation at 95 °C, 20 s primer annealing at 55 °C, 1 min extension at 72 °C, and a final 5 min extension at 72 °C. The PCR reaction mix was in a total volume of 30 μL, including 15 μL of 2×Es Taq MasterMix, 10.5 μL of ddH$_2$O, 1.5 μL of DNA template, and 1.5 μL of each primer [39]. The quality of the PCR products was checked by performing 1% agarose gel electrophoresis. DNA sequencing was performed using an ABI PRISM® 3730XL DNA analyzer with a BigDye®120 Terminator Kit v.3.1. The positive transformants were sequenced at Sangon Biological Engineering Co., Ltd. (Beijing, China) [46].

## 2.5. Phylogenetic Analysis

Sequences were assembled with the SeqMan program v.7.1.0 in DNASTAR Lasergene core suite software (DNASTAR Inc., Madison, WI, USA). Homologous sequences with high similarity from ex-type and non-type *Wilsonomyces*-like isolates were included to serve as phylogenetic references and obtained using the BLAST function in the National Center for Biotechnology (NCBI) database and extensive literature review. All sequences were subjected to Bayesian inference (BI) analysis using MrBayes 3.2.6 [47], maximum likelihood (ML) was performed using RAxML-HPC v.8 on XSEDE, and maximum parsimony (MP) analysis was performed using PAUP v. 4.0a150 [48–50]. Multiple sequences of concatenated ITS, LSU, and tef1 sequences were aligned using MAFFT v. 7 with default settings and edited manually using MEGA v.6.0. *Stigmina platani* was selected as an out-group in all analyses [50]. Phylogenetic trees were edited using FigTree v.1.4.0. The GenBank accession numbers for the sequences used in these analyses are given in Table 2 [24], and isolates collected for this study are reported in Table 1.

Bayesian inference (BI) analysis was performed using MrBayes v.3.2.6. Two independent analyses of two parallel runs and four chains were carried out for 5,000,000 generations and sampled every 5000 generations, resulting in 1000 trees in total. The first 25% of the resulting trees were eliminated as the burn-in phase of each analysis. Branches with significant BI posterior probability (BIPP) were estimated for the remaining 750 trees [51].

Maximum likelihood (ML) analysis was done using RAxML v.7.2.8 and a GTR (general-time-reversible, one of the most popular models of nucleotide substitution because it constitutes a good trade-off between mathematical tractability and biological reality) model of site substitution including estimation of gamma-distributed rate heterogeneity and a proportion of invariant sites [48]. The branch support was evaluated with the bootstrapping method with 1000 bootstrap replicates [52].

A maximum parsimony (MP) analysis was conducted using the heuristic search option of 1000 random-addition sequences with tree bisection and reconnection (TBR) branch swapping algorithm. Clade stability was assessed with a bootstrap analysis of 1000 replicates [52]. Descriptive tree statistics for parsimony tree length (TL), consistency index (CI), retention index (RI), rescaled consistency index (RC), and homoplasy index (HI) were calculated for the maximum parsimonious tree [51].

### 2.6. Pathogenicity Tests

Pathogenicity experiments were conducted in June 2018 to August 2019 in the Forest Pathology Laboratory of the College of Forestry and Horticulture, Xinjiang Agricultural University, Urumqi, China. In order to test for pathogenicity, young fruits and leaves were collected from healthy *P. divaricata* and *P. armeniaca* (wild apricots) from Xinyuan County of Ili, Xinjiang.

Leaves and fruits were inoculated using mycelial plugs (3 mm diameter). Inocula were prepared by growing individual fungal isolates on PDA at 25 °C in a 12 h photoperiod for 6 days. Healthy leaves and fruits of the same age and size were collected from trees, washed thoroughly with tap water, surface sterilized with 1% NaClO for 10 min, then washed with sterile distilled water three times, and placed on paper towels in the laminar flow cabinet for about 2 h. Inoculation was done following wounding with a sterile needle to create a single wound, which was then covered with a mycelial plug (3 mm) with the mycelium facing down, after being wrapped with parafilm. After 24 h, mycelial plugs were removed from the surface of the inoculated tissues. The control groups were mock inoculated with non-colonized PDA plugs. Inoculated leaves and fruits were placed on a grid tray inside a plastic crisper with water at the bottom and the lid closed to maintain high humidity. Crispers were incubated at room temperature at 25 ± 2 °C on a laboratory bench until development of symptoms. Each pathogenicity test used 10 replicates and all procedures were carried out under aseptic conditions. Pathogens were re-isolated from the resulting lesions and identified as described above [53]. Any changes in the tissues surrounding the inoculation sites on leaves and fruits were recorded over a period of 20 days. The length and width of lesions were measured using a vernier caliper (Mitutoyo 500-196; Mitutoyo). Based on the length and width, the oval area of lesions was obtained to indicate lesion size [54]. The data were analyzed by Microsoft Excel 2010 and SPSS 20.0 software. One-way ANOVA was used to determine significant differences in lesion size of different isolates and assess the pathogenicity of fungal isolates [55]. The treatment means were compared by Tukey's honest significant difference (HSD) test at $p = 0.05$ [56].

**Table 2.** Description of DNA sequences used in phylogenetic analysis.

| Species | Isolate no. | Collector | Host | Country | GenBank Accession Numbers | | |
|---|---|---|---|---|---|---|---|
| | | | | | ITS | LSU | tef1 |
| *Dothidotthia symphoricarpi* | CBS 119687[ET] | A. Ramaley | *Symphoricarpos rotundifolius* | USA | MH863064 | MH874618 | – |
| *Mycocentrospora acerina* | CBS 113.24 | A. van Luijk | *Carum carvi* | Netherlands | MH854764 | MH866268 | – |
| *Neoascochyta desmazieri* | CBS 247.79 | E. Lengauer | Gramineae | Austria | KT389507 | KT389725 | – |
| *Neoascochyta desmazieri* | CBS 297.69 | U.G. Schlösser | *Lolium perenne* | Germany | KT389508 | KT389726 | – |
| *Neoascochyta europaea* | CBS 819.84 | M. Hossfeld | *Hordeum vulgare* | Germany | KT389510 | KT389728 | – |
| *Neoascochyta europaea* | CBS 820.84 | M. Hossfeld | *Hordeum vulgare* | Germany | KT389511 | KT389729 | – |
| *Neoascochyta exitialis* | CBS 234.52 | E. Muller | *Triticum spelta* | Switzerland | MH857013 | MH868539 | – |
| *Neoascochyta exitialis* | CBS 316.81 | H.T. Jachmann | *Triticum aestivum* | Germany | MH861347 | MH873106 | – |
| *Phaeomycocentrospora cantuariensis* | CBS 132013 | H.D. Shin | *Acalypha australis* | South Korea | GU269667 | – | GU384384 |
| *Phaeomycocentrospora cantuariensis* | CBS 132014 | H.D. Shin | *Humulus japonicus* | South Korea | GU269668 | – | GU384385 |
| *Pleiochaeta carotae* | CPC 27452[T] | M. Truter | carrot | South Africa | KY905669 | KY905663 | – |
| *Pleiochaeta setosa* | CBS 118.25 | C.M. Doyer | Laburnum | – | KY929373 | KY929376 | – |
| *Pleiochaeta setosa* | CBS 502.80 | W. Gams | *Chamaespartium sagittale* | Austria | KY929374 | KY929377 | – |
| *Thyrostroma compactum* | CBS 335.37 | J.C. Carter | *Ulmus pumila* | – | KY905670 | KY905664 | KY905681 |
| *Thyrostroma cornicola* | CBS 141280[T] | P.W. Crous and H.D. Shin | *Cornus officinalis* | South Korea | KX228248 | KX228300 | KX228372 |
| *Thyrostroma franseriae* | CBS 487.71[T] | F.W. Went | *Franseria* sp. | USA | KX228249 | KX228301 | KY905680 |
| *Thyrostroma franseriae* | CBS 700.70 | F.W. Went | *Franseria* sp. | USA | KX228250 | MH871705 | KY905682 |
| *Wilsonomyces carpophilus* | CBS 231.89[ET] | J.W. Veenbaas-Rijks | *Prunus subhirtella* | – | KY905672 | KY905666 | KY905684 |
| *Wilsonomyces carpophilus* | CBS 159.51 | G. Goidanich | – | Italy | KY905671 | KY905665 | KY905683 |
| *Stigmina platani* | CBS 125773 | R. Zare | – | Iran | MH863752 | MH875220 | – |

CBS, Westerdijk Fungal Biodiversity Institute, Utrecht, the Netherlands; CPC, Culture Collection of Pedro Crous, housed at Westerdijk Fungal Biodiversity Institute. [T] and [ET] indicate ex-type and ex-epitype strains, respectively. ITS, internal transcribed spacers and intervening 5.8S nrDNA; LSU, 28S large subunit RNA gene; tef1: partial translation elongation factor 1-alpha gene.

## 3. Results

### 3.1. Field Symptoms and Isolating the Fungi

A total of 9600 leaves and 3200 fruits from 80 *P. armeniaca* trees were surveyed in Xinyuan, Yining, Huocheng, and Gongliu Counties; the average frequency of infection of leaves and fruits was 79.33% and 53.17%, respectively. Symptoms on leaves consisted of 1–2 mm circular spots, purplish in color with a yellow bordering halo, which eventually enlarged and became necrotic, causing abscission in the center of the lesion, giving the leaf the typical shot-hole appearance (Figure 1a). Symptoms on fruits appeared as raised brown sores or cracks, and the fruits were necrotic (Figure 1f).

A total of 800 fruits from 20 *P. divaricata* trees were surveyed in Huocheng County, with an infection rate of 37.45%. Symptoms on *P. divaricata* fruits appeared as round spots, dark brown in the center and yellowish brown on the edges, with a hardened peel (Figure 1k).

Three types of fungi (*Wilsonomyces*-like, *Didymella* sp., and *Alternaria* sp.) were obtained from 705 tissue blocks of 47 samples. No bacterial colonies were isolated from any samples. *Wilsonomyces*-like isolations collected from *P. armeniaca* leaves accounted for 72.8% of the total isolates. All isolates collected from fruits of *P. armeniaca* and *P. divaricata* were *Wilsonomyces*-like. Occasionally isolates of *Didymella* sp. and *Alternaria* sp. were also collected from leaves, accounting for 9.6% and 17.6%, respectively, of the total fungi isolated. They were not included in morphological, phylogenetic, and pathogenicity studies because of their low frequency. Pure growths of 25 *Wilsonomyces*-like isolates (three from *P. divaricata* fruits, 10 from *P. armeniaca* fruits, and 12 from *P. armeniaca* leaves) were selected for single-spore purification for this study.

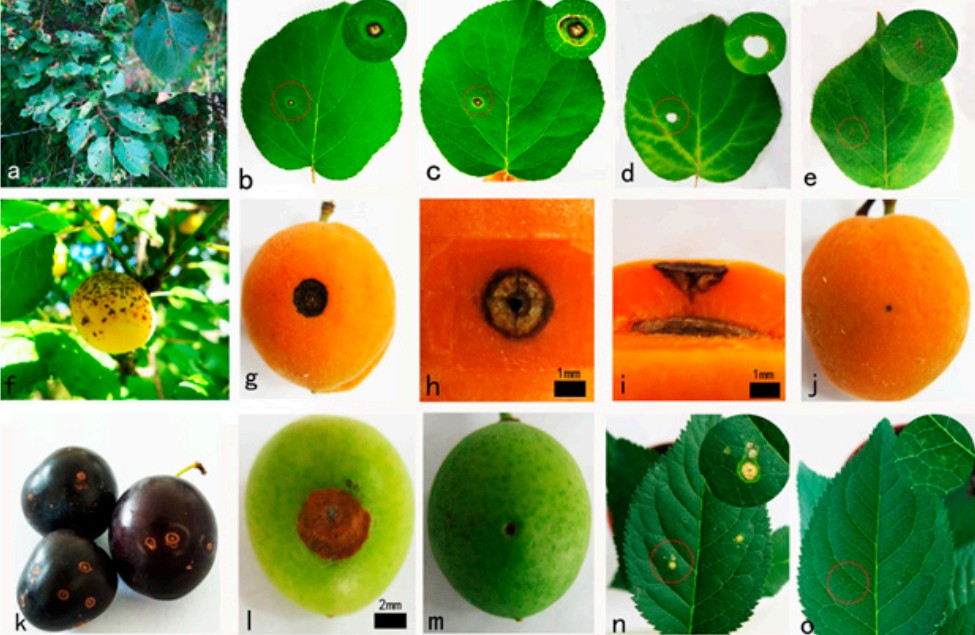

**Figure 1.** Symptoms in the field and inoculated *W. carpophilus* fungus symptom development during pathogenicity tests on *Prunus armeniaca* and *Prunus divaricate*: (**a**) natural disease symptoms on *P. armeniaca* leaves growing in the field; (**b–d**) inoculation on *P. armeniaca* leaves (3, 9, and 17 d); (**e**) *P. armeniaca* leaf control; (**f**) natural disease symptoms on *P. armeniaca* fruits growing in the field; (**g–i**) inoculation on *P. armeniaca* fruits for 12 d (h: transverse section through lesion; i: longitudinal section through lesion); (**j**) *P. armeniaca* fruit control; (**k**) natural disease symptoms on *P. divaricata* fruits growing in the field; (**l**) inoculation on *P. divaricata* fruits for 6 d; (**n**) inoculation on *P. divaricata* leaves for 10 d; (**m,o**) control with *P. divaricata* fruits and leaves. Scale bars: h, i = 1 mm; l = 2 mm.

### 3.2. Cultural and Morphological Characteristics

Strong variability in cultural and morphological characteristics was recorded among the different isolates of *Wilsonomyces*-like on PDA medium. The 25 isolates of the pathogen were divided into eight morphological groups based on the shape of the colony margin, concentric growth patterns, and color in PDA medium. Groups I–V were characterized by a regular colony margin, nearly round, and an average growth rate of 0.47 cm/d. Group I, including nine isolates (G048 3m3, G048 5m2, G048 7m1, G049 7m1, G053 5m1, Y039 3m3, Y046 7m2, Y049 7m1, Y045 5m2-2), was characterized by dull white to pale yellow colonies with no concentric ring pattern (Figure 2a); Group II included four isolates (G052 5m2, G052 5m3, G053 7m3,Y057 7m3) with gray to light yellow colonies with concentric ring patterns (Figure 2 b); Group III included four isolates (G004 5m2, G004 7m2, G010 5m2, G059 5m2) with light-yellow to brown colonies with no concentric ring pattern (Figure 2c); Group IV included a single isolate (Y037 7m2) with dull olivaceous to brown colonies and a concentric ring pattern (Figure 2d); Group V included three isolates (Y035 5m1, Y038 7m2, Y040 7m2) with brown to dark brown colonies with a concentric ring pattern (Figure 2e). Groups VI to VIII had irregular margins, no concentric ring pattern, and slow growth, averaging 0.21 cm/d. Group VI comprised two isolates (Y043 7m1, Y052 7m1) with dull white to dark gray colonies (Figure 2f); Group VII had one isolate (Y048 5m2) with dark olivaceous to black colonies (Figure 2g); Group VIII had one isolate (G048 5m3) showing dull white to pale yellow colonies (Figure 2h).

Three isolates obtained from *P. divaricata* fruits were classified into the morphological Group III. Twelve isolates from *P. armeniaca* leaves were classified into six morphological groups: I, II, IV, V, VI, and VII. Ten *P. armeniaca* fruit isolates were classified into four morphological groups: I, II, III, and VIII. Group I is the most common morphological type of *P. armeniaca*.

The shape and color of conidia were consistent among the isolate groups. Conidia were produced after 7 days in PDA medium and were most abundant at the edge and darkening part of the colony. Conidia were spindle-shaped, with ovate apical cells and truncate basal cells (2.11–4.74 µm), with 1–5 transverse septa, and with a majority of conidia comprising three septa, (18.07–)28.66–40.29(–56.39) × (10.54–)12.83–16.15(–17.37) µm, colorless and transparent when immature, turning sub-hyaline and dark olivaceous to dark brown with age (Figure 2i,j).

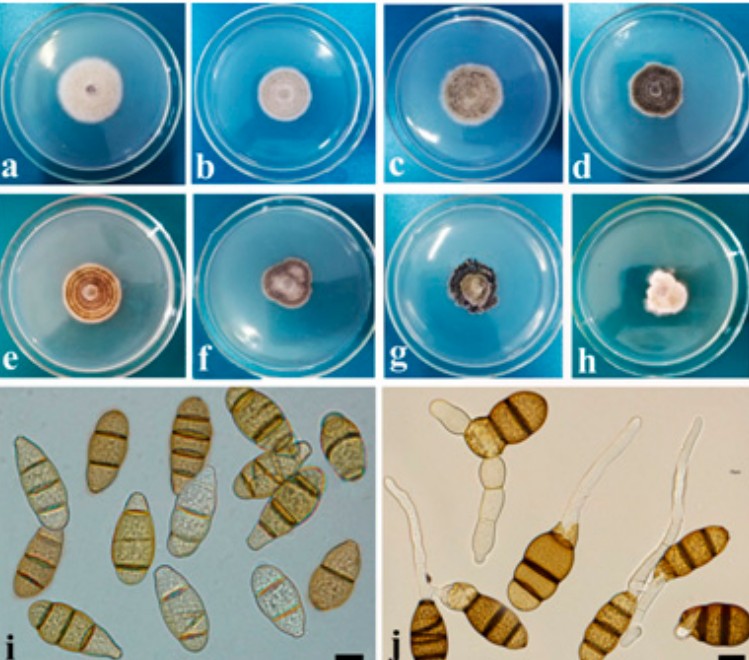

**Figure 2.** Colonies of *Wilsonomyces carpophilus* isolates and conidia characteristics: (**a–h**) Groups I to VIII of the culture (**a–e**: 10 d; **f–h**: 18 d); (**i**) conidia; (**j**) germinated spores. Scale bars: i, j = 10 µm.

### 3.3. Phylogenetic Analysis

The ITS sequence of 25 *Wilsonomyces*-like isolates showed high similarity to *Wilsonomyces carpophilus* reference sequences available in GenBank. The ITS, LSU, and tef1 sequences obtained from the 25 isolates in this study were submitted to GenBank (Table 1).

The ITS, LSU, and tef1 sequences from 44 taxa belonging to two families (genera of Dothidotthiaceae and Didymellaceae) were used in the phylogenetic analysis. The combined datasets comprised 2620 characters, of which 1724 characters were constant, 461 variable characters were parsimony uninformative, and 435 were parsimony informative. In the MP analysis, the reconstructed trees were 1367 steps long, with CI = 0.815, RI = 0.866, RC = 0.706, and HI = 0.185. BIPP, ML, and MP bootstrap support values above 50% are given above or below the branches (Figure 3). The three types of analysis trees resulting from the concatenated dataset showed the same relationships among *Wilsonomyces*-like isolates. The reference strains (CBS 159.51 and ex-epitype CBS 231.89) were clustered with the new isolates and had a higher support value (BI/ML/MP = 85/89/100) in Figure 3. The 25 *W. carpophilus* isolates also showed high genetic diversity in the phylogenetic analysis, and the phylogenetic groups did not correspond to the morphological groups.

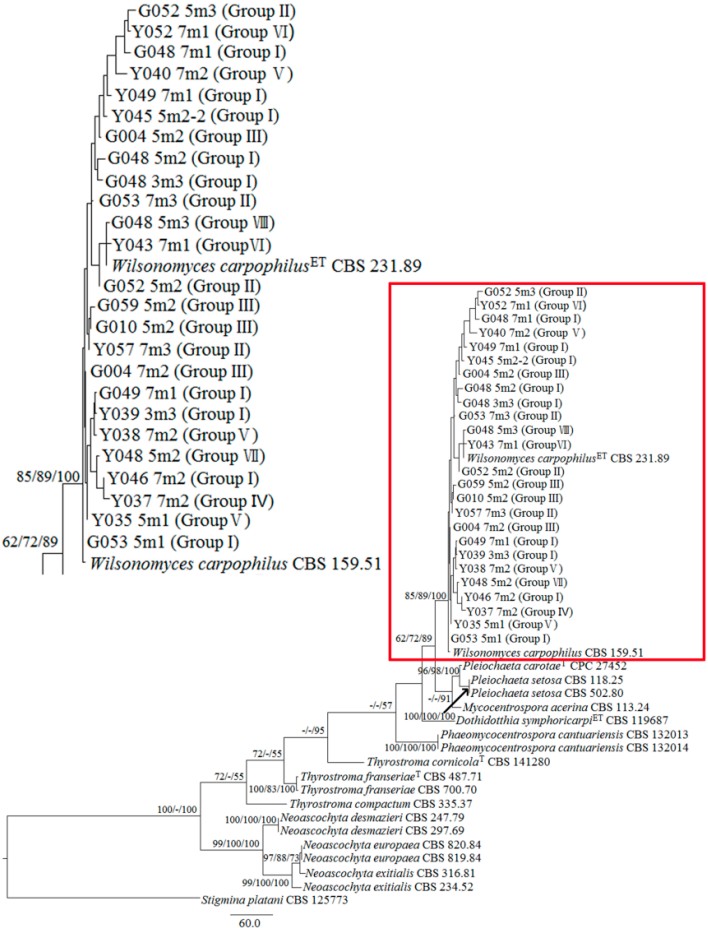

**Figure 3.** Phylogenetic tree showing phylogenetic relationships among species of Dothidotthiaceae inferred from a three-gene dataset comprising sequences of rDNA internal transcribed spacer (ITS) region, partial large subunit (LSU), and translation elongation factor 1-alpha (tef1) gene sequences. Backbone of the tree was constructed using Bayesian analysis. *Stigmina platani* was selected as an outgroup. Bootstrap percentages of Bayesian posterior probabilities, maximum-likelihood, and maximum parsimony from 1000 replicates are shown, from left to right, on the deep and major resolved branches. Morphological groups in this study were labeled after isolate numbers.

### 3.4. Pathogenicity Tests

The 25 isolates were used for pathogenicity tests. For inoculation of *P. armeniaca*, 12 isolates of *W. carpophilus* were inoculated on leaves, and 10 isolates of *W. carpophilus* were inoculated on fruits. All leaves inoculated with of *W. carpophilus* developed circular brown lesions with pale centers within 3–5 days after inoculation. Then, about 0.5 mm of the mesophyll tissue at the outer edge of the lesions became thinner, and the boundary between healthy and necrotic tissue became obvious. After 17 days post-inoculation, the whole lesion collapsed, and a small hole formed (Figure 1b–d). No significant difference in virulence was found among the 12 *W. carpophilus* isolates inoculated on leaves ($p = 0.346$) (Figure 4a), and the average lesion size was 5.38 mm$^2$.

Inoculated *P. armeniaca* fruits developed sunken necrotic lesions within 3–12 days after inoculation (Figure 1g–i). There was a very significant difference between strains G053 7m3 and G052 5m2, with lesions averaging 10.75 and 5.57 mm$^2$, respectively; the other eight inoculated strains showed no significant difference ($p < 0.0001$) (Figure 4b). Both leaves and fruits of *P. armeniaca* in the control group remained asymptomatic (Figure 1e,j). Lesions produced in the inoculated leaves and fruits were similar to symptoms observed in the field. *W. carpophilus* was re-isolated from symptomatic tissue in all inoculated leaves and fruits, thus fulfilling Koch's postulates.

For *P. divaricata*, three isolates of *W. carpophilus* from fruits were inoculated onto leaves and fruits, and lesions developed in all inoculated organs. Inoculated leaves developed circular brown lesions within 5–10 days after inoculation. Although the mesophyll tissue became thinner, the lesion did not detach to form a perforation (Figure 1n). There were no significant differences among the three isolates tested ($p = 0.072$) (Figure 4c). The average lesion size in leaves was 4.21 mm$^2$.

Inoculated fruits of *P. divaricata* developed brown necrotic lesions within 3–6 days after inoculation with *W. carpophilus* isolates (Figure 1l). The isolate G010 5m2 showed extremely significant differences with G004 7m2 and G004 5m2. There was no significant difference between G004 7m2 and G004 5m2 ($p = 0.014$) (Figure 4d). The average lesion size for inoculated G004 7m2, G004 5m2, and G010 5m2 was 20.02, 19.95, and 13.65 mm$^2$, respectively. Lesions produced in the inoculated fruits were similar to symptoms observed in the field. No lesions were found in the various control treatments (Figure 1m,o). *W. carpophilus* was also re-isolated from all symptomatic leaves and fruits, thus fulfilling Koch's postulates.

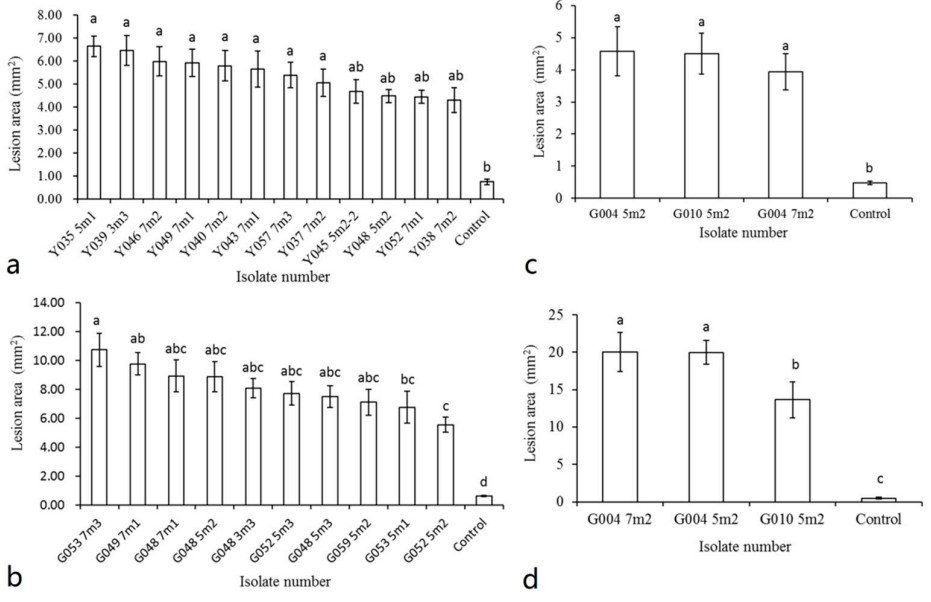

**Figure 4.** Mean (±SD) lesion area (mm$^2$) developed following inoculation of *Prunus armeniaca* and *Prunus divaricata* excised leaves and fruits with different isolates of *Wilsonomyces carpophilus*. Means that were significantly different based on honest significant difference (HSD) analysis are labeled with

different letters (*n* = 10, *p* = 0.05). (**a**) *P. armeniaca* leaves; (**b**) *P. armeniaca* fruits; (**c**) *P. divaricata* leaves; (**d**) *P. divaricata* fruits.

## 4. Discussion

Shot-hole disease affecting stone fruits in China has been historically attributed to the bacterium *Xanthomonas arboricola* pv. *pruni*. Bacterial spot was shown in eight peach-producing areas in China [57,58]. The shot-hole disease affecting stone fruits in the Western Tianshan Mountains was generally thought to be caused by *Xanthomonas arboricola* pv. *pruni*, but *P. armeniaca* fruit spot caused by *W. carpophilus* in Gongliu County, Xinjiang, was reported in 2019 [1,59]. Our study indicates that *W. carpophilus* not only infected *P. armeniaca* fruits, but also caused shot-hole disease in *P. divaricata* fruits and *P. armeniaca* leaves [59].

The shot-hole disease of stone fruits caused by *W. carpophilus* can be confused with the bacterial spot disease caused by *X. arboricola* pv. pruni, due to overall similar symptoms [60]. It is likely that the two diseases were confused in previous efforts to document diseases affecting the wild-fruit forest of the Western Tianshan Mountains. Therefore, lesion characteristics alone should not be used to diagnose fruit and leaf spot diseases of stone fruits, and isolation procedures should be conducted systematically for accurate disease diagnosis.

In this study, the conidia characteristics of the isolates were similar to those of *W. carpophilus*: conidia (18.07–)28.66–40.29(–56.39) × (10.54–)12.83–16.15(–17.37) μm vs. (27–)32–45(–55) × (12–)13–14(–16) μm, transversal septate (2–5), colorless and transparent when immature, turning sub-hyaline, dark olivaceous to dark brown with age [24]. We observed large differences in culture morphology between the 25 isolates of *W. carpophilus*. According to morphological characteristics such as the shape of the colony margin, concentric growth patterns, and color, eight morphological groups were identified. The isolates obtained from *P. divaricata* fruits had consistent morphology and were classified into Group III. Strains derived from *P. armeniaca* leaves and fruits showed the highest variation in colony morphology. Strains derived from *P. armeniaca* fruit were divided into four groups, and strains derived from *P. armeniaca* leaves were divided into six groups, and the majority of *P. armeniaca* strains were included in Group I. However, the *P. armeniaca* strains could not be grouped according to either their geographical origin or the host tissue from which they were isolated. Although there were differences in culture morphology, no significant differences in color and size of the conidia were found among these isolates, indicating that these variations are more pronounced at the species level. These results agree with those reported by Nabi et al., who determined significant variations of morphology, particularly cultural characteristics, of *W. carpophilus* isolates from stone fruits in India, and suggested that the nonsignificant variation in conidial shape and color recorded in different isolates explained that these variations are much more pronounced at the species level and less pronounced at intra-species levels [61]. Studies of other pathogens (*Magnaporthe grisea*, *Colletotrichum gloeosporioides*, and *Colletotrichum capsici*) also indicate that conidial shape and color are significant only at the species level only [62,63].

Similarly, phylogenetic analysis shows an obvious clustering of *W. carpophilus* and the 25 isolates in this study into a single branch with higher support value. The *W. carpophilus* isolates also had high genetic diversity. However, morphological groups did not correspond to phylogenetic groups, despite some genetic variation among *W. carpophilus* isolates. Different *W. carpophilus* isolates have been previously reported to have high intraspecific genetic diversity. Ahmadpour et al. showed high genetic diversity in 28 isolates of *W. carpophilus* from different regions in Iran using DNA fingerprinting by random amplified polymorphic DNA polymerase chain reaction (RAPD-PCR) and four random primers [64]. A high level of polymorphism in different isolates of *W. carpophilus* in Kashmir using seven ISSR (inter-simple sequence repeat) markers also indicated that these markers were suitable for studying the genetic diversity in shot-hole pathogens [61]. The 25 isolate sequences in this study showed high similarity with *W. carpophilus* reference sequences available from GenBank. *Wilsonomyces* is a monotypic genus. *Helminthosporium carpophilum* was initially described and transferred to different genera until Adaskaveg et al. introduced *Wilsonomyces* to accommodate this species [21]. Sutton regarded *W. carpophilus* as a synonym of *Thyrostroma* [25]. However, Marin-Felix

et al. separated these last species based on LSU, ITS, and tef1 sequence analysis and introduced *Thyrostroma compactum*, confirming that *Wilsonomyces* represents a distinct genus belonging to the Dothidotthiaceae [24]. In this study, the results of molecular phylogenetic analysis using multiple genes are consistent with those of Marin-Felix et al.

Isolates of *W. carpophilus* obtained from fruits of *P. divaricata* showed pathogenicity in fruits and leaves of *P. divaricata*. Isolates obtained from *P. armeniaca* leaves and fruits also were pathogenic to this host. Analysis of the morphological characteristics and gene sequences confirmed that strains re-isolated from all lesions produced in the pathogenicity study were identical to the inoculated ones, thereby fulfilling Koch's postulates. The pathogenicity of *P. divaricata* and *P. armeniaca* determined that there were no significant differences among different isolates inoculated on *P. divaricata* and *P. armeniaca* leaves ($p < 0.05$). There were significant differences in inoculation on *P. divaricata* and *P. armeniaca* fruits ($p > 0.05$). The symptoms produced on the detached fruits were bigger than those on the fruits in the field, which may be due to the wound, the amount of fungus inoculated, or the influence of environmental factors. Nabi et al. observed that *W. carpophilus* produced more severe lesions on injured than uninjured tissues, and the isolates varied significantly with respect to the incubation period and the subsequent size (diameter) of the lesions produced [61].

## 5. Conclusions

This study presents the first research on the serious shot-hole disease on *P. divaricata* and *P. armeniaca* (wild apricot) leaves caused by *W. carpophilus* in the wild-fruit forest of the Western Tianshan Mountains. Twenty-five pure isolates of *W. carpophilus* were selected from *P. armeniaca* leaves and fruits and *P. divaricata* fruits. Morphological characteristics, phylogenetic analysis, and pathogenicity were recorded for all strains. All re-isolated strains from all lesions were identified, thereby fulfilling Koch's postulates. According to the morphological characteristics, the 25 isolates were divided into eight groups. The pathogenicity tests also showed significant differences in some strains on inoculated fruits. This study has crucial implications for shot-hole disease diagnosis and pathogen detection. We also suggest that the genetic diversity of *W. carpophilus* from the wild-fruit forest needs further study.

**Author Contributions:** All authors read and agree to the published version of the manuscript. Conceptualization, C. T. and R. M.; Formal analysis, S. Y., R. M. and G. C.; Funding acquisition, R. M.; Investigation, R. M., S. Y. and H. J.; Software, G. C.; Visualization, S. Y.; Writing—original draft, S. Y., R. M.; Writing—review and editing, R. M., Dr. F. P. T.. All authors have read and agreed to the published version of the manuscript.

**Funding:** This research was funded by Projects Supported by the Natural Science Foundation of Xinjiang Uygur Autonomous Region, China, grant number 2018D01A19.

**Acknowledgments:** We want to thank Florent P. Trouillas, a researcher at the University of California, Davis, and Xiangyang Shi, who works in the United States Department of Agriculture's Agricultural Research Service, for modification of this article. We are also grateful to the cooperation projects (grant number G20190246001). We would also like to thank the Forest Inspection Bureau of Ili Prefecture for the convenience of this disease investigation and the Forest Inspection Bureau of Xinyuan County for providing materials for pathogenesis experiments. At the same time, we also wish to thank Liqiang Liu, Kang Liao, Ying Zhao, and Min Wang, who provided help and support during the experiment.

**Conflicts of Interest:** The authors declare no conflict of interest.

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
