# Peer review of "Morphology, DNA Phylogeny, and Pathogenicity of Wilsonomyces carpophilus Isolate Causing Shot-Hole Disease of Prunus divaricata and Prunus armeniaca in Wild-Fruit Forest of Western Tianshan Mountains, China"

_forests, doi:10.3390/f11030319_

Round 1

Reviewer 1 Report

Shanghua et al. provide an interesting and well-designed study characterising the effect of a pathogenic fungus, Wilsonomyces carpophilus, on wild fruit trees. I would recommend this paper for publication, subject to the improvements recommended below:

Overall, the English is understandable, but in places it is unclear. I would recommend editing by a native speaker.

30-31: At first glance, this is confusing – I suggest it would be clearer if the sentence starts “On fruit, G053 7m3 and G052 5m2 were significantly different to each other on P. armeniaca….”

31: I presume you mean P > 0.05

32: I presume you mean P < 0.05

73: You previously said that this is the first report of W. carpophilus causing shot-hole disease in China, but here you indicate it has already been recorded – please check which statement is correct.

74: I presume you mean “Cases have been recorded on P. divaricata  only in….”

76-77: this is not a sentence.

76-87: this paragraph needs working in a more logical order. At present it jumps around between nomenclature, phylogeny and morphology.

106-108: were only infected trees sampled? Also, how did you distinguish between fungal and bacterial shot-hole disease?

149-150: please provide sequences for the primers

224: leaves and fruits

294-295: this isn’t clear – it could be interpreted as saying the reference strains clustered together, separately from the new isolates

Fig 3: it would be nice to see the Wilsonomyces strains in more detail. Would it be possible zoom in further on this part of the tree?

315-326: were isolates only inoculated onto the species they were originally found on?

345-346: does this data mean that Wilsonomyces accounts for most of the shot-hole cases, rather than the bacterial agent they were previously attributed to? Line 224 of your results indicates an astonishingly high prevalence.

Author Response

Response to Reviewer 1 Comments

Point 1: Overall, the English is understandable, but in places it is unclear. I would recommend editing by a native speaker.

Response 1: Thank you for your suggestion, we have did it by MDPI.

Point 2: 30-31: At first glance, this is confusing-I suggest it would be clearer if the sentence starts “On fruit, G053 7m3 and G052 5m2 were significantly different to each other on P. armeniaca….”

Response 2: Thank you for your suggestion. Be revised as: On fruit, G053 7m3 and G052 5m2 showed significant differences in inoculation on P. armeniaca, and G010 5m2 showed extremely significant differences with G004 7m2 and G004 5m2 on P. divaricata.

Point 3: 31: I presume you mean P > 0.05

Response 3: Thank you for your suggestion. Yes, it has been modified to P > 0.05.

Point 4: 32: I presume you mean P < 0.05

Response 4: Thank you for your suggestion. Yes, it has been modified to P < 0.05.

Point 5: 73: You previously said that this is the first report of W. carpophilus causing shot-hole disease in China, but here you indicate it has already been recorded – please check which statement is correct.

Response 5: Thank you for your suggestion. The wild species of P. armeniaca referred to in our study, which is indeed the first reported on wild apricot leaves in China. In line 73, the P. armeniaca is widely cultivated. They are different from each other, despite having the same Latin scientific name. To avoid confusion, change the last sentence of the abstract to "This is the first reports the shot-hole disease of P. armeniaca (wild apricot) leaves and P. divaricata occurred by W. carpophilus in China "

Point 6: 74: I presume you mean “Cases have been recorded on P. divaricata only in….”

Response 6: Thank you for your suggestion. Be revised as: Cases have been recorded on P. divaricata only in California and Poland (https://nt.ars-grin.gov/ fungaldatabas-es/), it has not been recorded in China.

Point 7: 76-77: this is not a sentence.

Response 7: Thank you for your suggestion. Be revised as: Shot-hole disease of stone fruits is caused by Wilsonomyces carpophilus (Lev.) Adask. (Bas. Helminthosporium carpophilum. syn. Stigmina carpophila, Coryneum beijerinkii, Clasterosporium carpophilum, Thyrostroma carpophilum, Sciniatosporium carpophilum, and Sporocadus carpophilus), an anamorph of the genus Wilsonomyces.

Point 8: 76-87: this paragraph needs working in a more logical order. At present it jumps around between nomenclature, phylogeny and morphology.

Response 8: Thank you for your suggestion. The sequence of this paragraph has been adjusted to the nomenclature, phylogeny and morphology of the fungus. The order of references has been adjusted accordingly.

Point 9: 106-108: were only infected trees sampled? Also, how did you distinguish between fungal and bacterial shot-hole disease?

Response 9: Thank you for your suggestion. Yes, we only infected the trees sampled. In current study, we make sure the pathogen which cause the shot-hole disease of Prunus divaricata is the first important thing. Next, we will do infect the other trees in the Wild-Fruit Forest.

Bacterial shot-hole disease just has been recorded in previous reports. In our study, isolation plates were checked the bacterial pathogen Xanthomonas arboricola pv. pruni. But we did not found it.

Point 10: 149-150: please provide sequences for the primers

Response 10: Thank you for your suggestion. Be revised as: Primers used were ITS1 (5'- TCC GTA GGT GAA CCT GCG G -3') and ITS4 (5'- TCC TCC GCT TAT TGA TAT GC -3') for ITS [42], NL1 (5'- GCA TAT CAA TAA GCG GAG GAA AAG -3') and NL4 (5'- GGT CCG TGT TTC AAG ACG G -3') for LSU [43] and EF1-688F (5'- CGG TCA CTT GAT CTA CAA GTG C -3') and EF1-1251R (5'- CCT CGA ACT CAC CAG TAC CG -3') for tef1[44].

Point 11: 224: leaves and fruits

Response 11: Thank you for your suggestion. We change to: A total of 9600 leaves and 3200 fruits from 80 P. armeniaca trees; A total of 800 fruits from 20 P. divaricata trees.

Point 12: 294-295: this isn’t clear-it could be interpreted as saying the reference strains clustered together, separately from the new isolates

Response 12: Modified this sentence to “The reference strains (CBS 159.51 and ex-epitype CBS 231.89) were clustered with the new isolates and had a higher support value (BI/ML/MP=85/89/100) in Figure 3.”

Point 13: Fig 3: it would be nice to see the Wilsonomyces strains in more detail. Would it be possible zoom in further on this part of the tree?

Response 13: Thank you for your suggestion. We had re-edited the tree to make it clearer.

Point 14: 315-326: were isolates only inoculated onto the species they were originally found on?

Response 14: Yes, all isolates had only inoculated onto the species they were originally found.

Point 15: 345-346: does this data mean that Wilsonomyces accounts for most of the shot-hole cases, rather than the bacterial agent they were previously attributed to? Line 224 of your results indicates an astonishingly high prevalence.

Response 15: In our study, isolation plates were checked the bacterial pathogen Xanthomonas arboricola pv. pruni. But we did not found it. And W. carpophilus was re-isolated from symptomatic in all the inoculated leaves and fruits, thus completing Koch’s postulates.

Reviewer 2 Report

The manuscript forests-715849 by Shuanghua & al study the etiology of a shot-hole disease of wild Prunus species in western China. They clearly show that the disease is caused by a fungus, Wilsonomyces carpophilus, by isolating, and characterising the fungus from 5 locations in China and by fullfilling the Koch postulates. Despite quite high morphological variability, the authors demonstrate without doubt that all collected isolates belong to W. carpophilus.The manuscript is generally well structured and written, the objectives are clearly stated and are met by the results. The methodology applied is correct. I thus would recommend the publication of this manuscript in Forests.

I just have some minor remarks

  • It is suggested in the introduction that this disease could be connected to a dieback of the wild Prunus species involved, P. divaricata (L64). However little inforlmation is given. It appears strange to me that a shot-hole disease may reach such a severity. More precision to the relation with this decline could be given. Dies the shot-hole induce severe defoliation?
  • The english of the manuscript should be corrected as there are numerous small grammatical errors, although it remains easy to understand. Not being a native english I do not propose corrections. Some sentence are difficult to understand and should be rewritten (L366-370, L378-380).
  • L128-140. I guess the 25 isolates selected for morphological and molecular analysis are the same. The authors may want to tell that more clearly. Also, thae fact that they were selected to encompass all the morphological variability observed in the sampling is told just L141. It would be more logical to put is before L130.

Author Response

Response to Reviewer 2 Comments

Point 1: It is suggested in the introduction that this disease could be connected to a dieback of the wild Prunus species involved, P. divaricata (L64). However little information is given. It appears strange to me that a shot-hole disease may reach such a severity. More precision to the relation with this decline could be given. Dies the shot-hole induce severe defoliation?

Response 1: The symptoms on the leaves include small circular purple lesions with pale canters, which gradually enlarged and became necrotic in the centre until the centre fell out, giving the appearance of shot-hole. Leaf photosynthesis will be seriously affected. On the fruits, the pathogen caused sunken necrotic lesions with purplish halos. The growth of seeds and the regeneration of Prunus divaricata population also can be affected.

Point 2: The english of the manuscript should be corrected as there are numerous small grammatical errors, although it remains easy to understand. Not being a native english I do not propose corrections. Some sentence are difficult to understand and should be rewritten (L366-370, L378-380).

Response 2: Thank you for your suggestion, we have did it by MDPI.

Point 3: L128-140. I guess the 25 isolates selected for morphological and molecular analysis are the same. The authors may want to tell that more clearly. Also, the fact that they were selected to encompass all the morphological variability observed in the sampling is told just L141. It would be more logical to put is before L130.

Response 3: Thank you for your suggestion. Adjust “Twenty-five isolates representing various morphological groups (3 from P. divaricata fruits, 10 from P. armeniaca fruits, and 12 from P. armeniaca leaves) were used for morphological identification.” to 130 lines before.